# Excitation of Hybrid Waveguide-Bloch Surface States with Bi_2_Se_3_ Plasmonic Material in the Near-Infrared Range

**DOI:** 10.3390/mi13071020

**Published:** 2022-06-28

**Authors:** Hongjing Li, Gaige Zheng

**Affiliations:** 1School of Electronics Engineering, Nanjing Xiaozhuang University, Nanjing 211171, China; hongjingli@njxzc.edu.cn; 2Jiangsu Collaborative Innovation Center on Atmospheric Environment and Equipment Technology (CICAEET), Nanjing University of Information Science & Technology, Nanjing 210044, China

**Keywords:** Bloch surface waves, Bi_2_Se_3_, distributed Bragg reflector

## Abstract

Bloch surface waves (BSWs) with Bi_2_Se_3_ in a composite structure consisting of a coupling prism, distributed Bragg reflector (DBR) and cavity layer have been demonstrated. The design relies on the confinement of surface waves that originates from the coupling between the defective layer of plasmonic material (Bi_2_Se_3_) and DBR. The presence of the cavity layer modifies the local effective refractive index, enabling direct manipulation of the BSWs. The transfer matrix method (TMM) is used to evaluate the reflectance and absorptance responses in the spectral domain for various angles of incidence, demonstrating the presence of sharp resonances associated with the BSW. With an optimal thickness of DBR bilayers, the energy of an evanescent wave can be transferred into the periodic stack resulting in the excitation of waveguide modes (WGMs). It is believed that the proposed design possesses the advantage in terms of easy fabrication to develop integrated photonic systems, especially for biological and chemical sensing.

## 1. Introduction

During the past decades, significant attention has been paid to platforms that can support electromagnetic surface waves (EMSWs) [1,2,3,4]. Among the various methods and structures, BSWs are the modes propagating at the interface between the dielectric multilayer and external medium decay exponentially inside the multilayers due to the presence of the photonic band gap (PBG). The BSW field distributions can be tuned by tailoring the thickness and material of the topmost layer, which enables the investigation of interactions with external media. BSW-based sensors have been proved can provide several advantages compared to surface plasmon polaritons (SPPs)-based sensors [5,6,7]. Firstly, BSWs can be excited under both polarizations and any wavelength by changing the composition and parameters of the DBR suitably. In addition, BSWs exhibit much sharper resonances than conventional surface plasmon resonance (SPR) [8,9]. Moreover, the multilayer structure can be fabricated without any pattern, making BSW an attractive optical sensing tool [10,11,12].

With the preparation and development of new materials, the performance of optical devices can be significantly improved [13,14,15,16,17,18]. Tetradymites have become popular due to the recently discovered topological insulator property [19,20]. Bismuth selenide (Bi_2_Se_3_) has been shown as one of the potential candidates for surface plasmons [21,22], which holds great potential in high-performance photonic devices [23,24,25]. At the same time, a Bi_2_Se_3_ topological insulator is easy to prepare at a low cost and can even achieve saturable absorption because of its unique physical properties [18]. Thus far, most of the research are concerned with realizing narrow and tunable resonance based on BSW, but it is rarely reported that the tunable and multichannel BSWs can be achieved simultaneously. Moreover, a novel Bi_2_Se_3_ platform for the excitation of BSWs is significant for two-dimensional integrated photonics.

The presented work shows the possibility of using a Bi_2_Se_3_–DBR composite structure for the excitation of BSWs. Angle dependence of the reflection and absorption spectra is studied through TMM. It is also found that the BSW response is strongly dependent on the thickness of the Bragg mirror layer, incident angle and refractive index of the prism. In addition, manipulating the reflection dips can be achieved by changing the thickness of the cavity layer. More importantly, the proposed design possesses the advantage in terms of easy fabrication and planar multilayer system, which holds great potential in efficient narrowband filters as well as beam splitters and deflectors.

## 2. Materials and Methods

The sample used for the BSW excitation consisted of a DBR and a Si cavity layer with a thin Bi_2_Se_3_ layer deposited on the top. As depicted in Figure 1, the DBR is composed of *N* pairs of alternately stacked TiO_2_ (~96 nm) and SiO_2_ (~172 nm) thin films deposited onto a prism substrate by means of ion beam sputtering. The incident light impinges on the structure from the prism with an angle of *θ*. *d*_0_, *d*_1_, *d*_2_, and *t* stand for the thicknesses of Bi_2_Se_3_, TiO_2_, SiO_2_, and cavity layer Si, respectively. The period number of TiO_2_/SiO_2_ Bragg mirror is set as *N*. At the wavelengths of interest, the refractive index for Si is fixed at 3.45.

Wavelength-dependent refractive indices for TiO_2_ [26] and SiO_2_ [27] as the high and low refractive index materials at a certain wavelength *λ* are given by:(1)$$n_{{\mathrm{TiO}}_{2}}=\sqrt{5.931+\frac{0.2441}{\lambda^{2}-0.0803}}$$
(2)$$n_{{\mathrm{TiO}}_{2}}=\sqrt{1+\frac{0.6962\lambda^{2}}{\lambda^{2}-0.0684^{2}}+\frac{0.4079\lambda^{2}}{\lambda^{2}-0.1162^{2}}+\frac{0.8975\lambda^{2}}{\lambda^{2}-9.8961^{2}}}$$

The number of periods for DBR is *N*; the entire device is supposed to be fabricated on a prism substrate. The monochromatic plane wave with an electric field parallel to the *y*-axis is called transverse electric (TE) polarization. Bi_2_Se_3_ has attracted great attention due to its hyperbolic behavior in the near-infrared to visible spectrum [21]. Accurately modeling the permittivity is vital for evaluating the optical response in Bi_2_Se_3_-based optoelectronic applications. The real and imaginary parts of the dielectric function on an *ac*-facet and *ab*-facet are obtained from Ref. [21], which is shown in Figure 2.

We simulate plane wave interactions with such planar multi-layer structures by using the semi-analytical transfer matrix method (TMM) [14,28,29]:(3)$$E_{out}=ME_{in}$$
where *E_out_* is the electric field of the output side, *E_in_* is the electric field of the incident light, and **M** is the transfer matrix of the whole structure, which can be expressed as follows:(4)$$M=\left(\begin{array}{cc} \begin{array}{l} M_{11}\\ M_{21} \end{array} & \begin{array}{l} M_{12}\\ M_{22} \end{array} \end{array}\right)={\left(m_{{\mathrm{TiO}}_{2}}m_{{\mathrm{SiO}}_{2}}\right)}^{N}m_{\mathrm{Si}}m_{{\mathrm{Bi}}_{2}{\mathrm{Se}}_{3}}$$
where $m_{{\mathrm{TiO}}_{2}}$, $m_{{\mathrm{SiO}}_{2}}$, *m*_Si_ and $m_{{\mathrm{Bi}}_{2}{\mathrm{Se}}_{3}}$ are the transmission matrix of light passing through each layer, and *N* is the period number of DBR.

The transmission matrix of incident light in each layer can be expressed as:(5)$$m_{i}=\left[\begin{array}{cc} \begin{array}{l} \cos\delta_{i}\\ -ip_{i}\sin\delta_{i} \end{array} & \begin{array}{l} -ip_{i}{}^{-1}\sin\delta_{i}\\ \cos\delta_{i} \end{array} \end{array}\right]$$
where *δ_i_* = (2*π*/*λ*)*n_i_d_i_*cos*θ_i_*, *n_i_d_i_* is the optical thickness of the corresponding layer, *θ_i_* is the angle between the light in the dielectric layer and the normal direction of the interface, and *λ* is the wavelength of the incident light. *p_i_* has the following formulas for TE and TM polarization, respectively:(6)$$p_{iTE}=n_{i}\sqrt{\varepsilon_{0}\mu_{0}}\cos\theta_{i}$$
(7)$$p_{iTM}=n_{i}\sqrt{\varepsilon_{0}\mu_{0}}/\cos\theta_{i}$$
where *i* = $m_{{\mathrm{TiO}}_{2}}$, $m_{{\mathrm{SiO}}_{2}}$, *m*_Si_ or $m_{{\mathrm{Bi}}_{2}{\mathrm{Se}}_{3}}$, *ε*_0_ and *μ*_0_ are the permeability of vacuum dielectric constant and vacuum dielectric constant, respectively.

The reflection (*R*) and the transmission (*T*) of light can be expressed as:(8)$$R={\left|\frac{\left(M_{11}+M_{12}\right)p_{i}-\left(M_{21}+M_{22}p_{T}\right)}{\left(M_{11}+M_{12}\right)p_{i}+\left(M_{21}+M_{22}p_{T}\right)}\right|}^{2}$$
(9)$$T={\left|\frac{2p_{i}}{\left(M_{11}+M_{12}p_{T}\right)p_{i}+\left(M_{21}+M_{22}p_{T}\right)}\right|}^{2}$$
where subscript *i* and *T* represent the incident and transmission space, respectively. The absorbance can be deduced through *A* = 1 − *R* − *T*.

## 3. Results and Analysis

A possibility of the excitation of surface waves using an evanescent field in a DBR geometry is highly favorable. We first study the Bi_2_Se_3_/DBR integrated with a prism to examine the role of BSW resonance. Figure 3a plots the optical responses (absorption and reflection spectra) of the hybrid DBR system consisting of three pairs (*N* = 3) of TiO_2_/SiO_2_. The conditions of Bragg conditions are met by selecting a quarter optical thickness of the DBR resonant wavelength for TiO_2_ and SiO_2_ layers. The thickness of the Bi_2_Se_3_ layer is 25 nm, and Si is chosen with a thickness of 80 nm as the cavity. It is clear that a deep reflection dip appears under 62.6° at the resonance wavelength of 1000 nm), indicating the BSW excitation at the Bi_2_Se_3_/air interface. With the increase of the number of periodic layers *N*, the absorption peaks of the structure gradually change, as shown in Figure 3b–d. When the angle of incident light is greater than the total reflection critical angle, an evanescent wave arising from total internal reflection (TIR) is formed. With an optimal thickness of TiO_2_/SiO_2_ bilayers, the energy of an evanescent wave can be transferred into the periodic stack resulting in the excitation of waveguide modes (WGMs). Thus, when the number *N* increases from three to nine pairs continuously, more absorption peaks appear.

Figure 4a–d show the reflection and absorption spectra with different *N* for TE polarization. It is evident that more resonance will appear with the increasing *N*. Reflection dips have been observed and are considered as the WGMs, and the incident waves are well localized within the DBR mirror. The different dips in the spectrum are related to the different order of guiding modes inside the multilayers. In the following study, we will restrict the study to a TE polarization monochromatic plane wave.

In addition to the number of DBR pairs, the dependence of optical reflection on the refractive index of the prism is also investigated, as depicted in Figure 5a,b, respectively. It is a key point that our design can achieve TIR, and the critical angle can be calculated $\theta_{c}=\arcsin\left(n_{eff}/n_{p}\right)$ according to Snell’s law. *n_eff_* is defined as the effective refractive index of TiO_2_/SiO_2_ pairs. Figure 5a shows the reflectance spectra of the structure for the TE waves at different *n_p_* and angles *θ* of incidence of the radiation onto the structure. It is evident that with a larger refractive index, more resonance can occur with the blue shift of the resonance angles. The reflectivity spectrum undergoes four pronounced reflection dips at 39.23°, 47.65°, 54.92° and 59.89°, which suggests that multiple BSW states are excited simultaneously when *n_p_* = 2.45 is in the near-infrared region.

To explore the physics mechanism of the existence of BSW states, a comprehensive reflectivity map, which is on the plane of incident angle and thickness of cavity layer (*t*), is illustrated in Figure 6. It is interesting to observe that the resonance of BSW is continuously variable with *t*, and the interval between the two neighboring stripes strictly equals *λ*/2*n*_Si_ for a given wavelength *λ*.

The propagation properties of the BSW mode on the platform can be altered by changing the thickness of the DBR layer. Their thicknesses define the location of the dispersion line in the band gap of the multilayers. Variation of reflection in the proposed geometry as a function of *d*_1_ and *d*_2_ when *λ*_central_ = 1 μm. For the tunability of the working wavelength, we break the constraint that the optical thickness of the SiO_2_ and TiO_2_ layer is a quarter of *λ*_central_. It is clear from Figure 7a that, with a fixed *d*_1_ (96 nm), as *d*_2_ increases to 800 nm, the device shows two bands of resonance with minimum reflection. This multiband response of the device increases the flexibility of the device preparation at a targeted wavelength. With a fixed *d*_2_ (172 nm), as *d*_1_ increases from 0.25 μm to 1.25 μm, multiple band gaps appear in the reflection map (shown in Figure 7b). Overall, it is convenient for one to obtain targeted optical responses by adjusting the thicknesses of planar films in DBR.

In Figure 8, the numerical experimental angle-resolved reflection spectra in the wavelength range 0.4–1.4 μm and incident angle 0–90° are shown for different prism refractive indices. We observe several optical modes that indicate the presence and location of a PBG of the multilayer structure. The BSW modes are excited because of the surface defect. Although such modes are intrinsically present at Bi_2_Se_3_–air interfaces, they are non-radiative in nature, as their momentum is larger than the free-space wave momentum [30,31]. Thus, we use a prism to provide additional momentum to the incident wave to excite BSWs.

## 4. Conclusions

In summary, we have demonstrated an angle-interrogated BSW device structure by exploiting the surface properties by introducing a Bi_2_Se_3_ layer on top of a prism-coupled photonic crystal. Comprehensive investigations regarding the effects of the structural parameters on the reflection and absorption properties have been conducted. The number of periodic layers *N* plays an important role in the occurrence of multiple BSWs in this structure. It was also found that the BSW response is strongly dependent on the thickness of the cavity layer, Bragg mirror layer, light incident angle and refractive index of the prism. Compared with other BSW-based devices of different types, it is shown that the proposed interfaces can support bound surface modes that differ significantly from the surface plasmon polaritons. These devices can be applied as efficient narrowband filters, beam splitters, and deflectors and also extend the usage of Bi_2_Se_3_.

## Figures and Tables

**Figure 1 micromachines-13-01020-f001:**
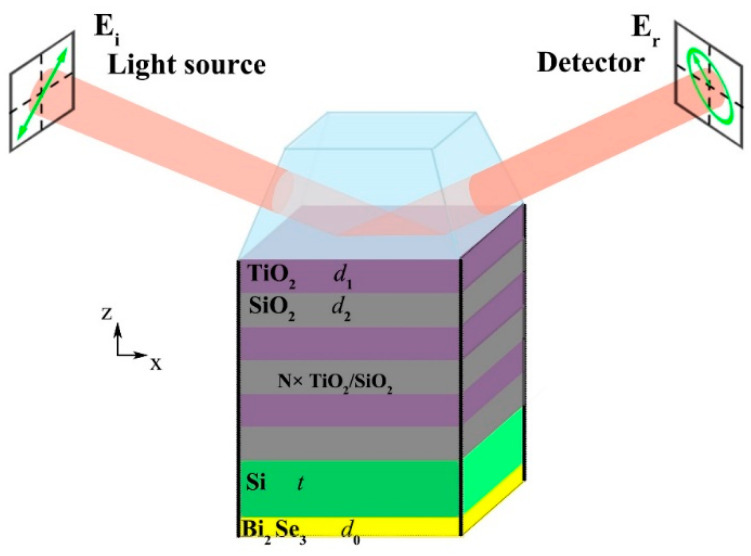
Schematic and structure parameters of the planar Bi_2_Se_3_/DBR integrated with a prism.

**Figure 2 micromachines-13-01020-f002:**
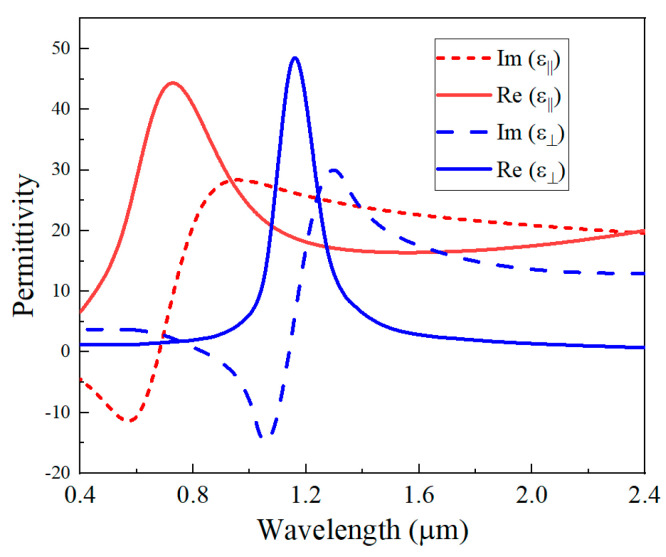
Imaginary and real parts of relative permittivity retrieved from generalized spectroscopic ellipsometry measurements of Bi_2_Se_3_ with *ac*-facet and *ab*-facet.

**Figure 3 micromachines-13-01020-f003:**
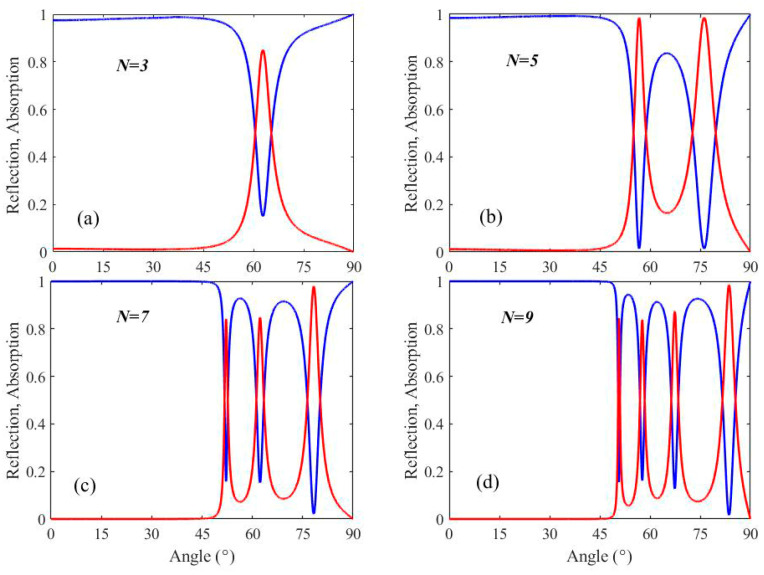
Absorption and reflection spectra of the DBR conjugated with the Bi_2_Se_3_ layer at a different period number of TiO_2_/SiO_2_ Bragg mirror under TE polarization. (**a**) *N* = 3, (**b**) *N* = 5, (**c**) *N* = 7 and (**d**) *N* = 9. The Bi_2_Se_3_ layer thickness is *d*_0_ = 25 nm, and the other parameters are set as *d*_1_ = 96 nm, *d*_2_ = 172 nm and *t* = 80 nm. The SF10 prism is used as the matrix (with refractive index *n_p_* = 1.7).

**Figure 4 micromachines-13-01020-f004:**
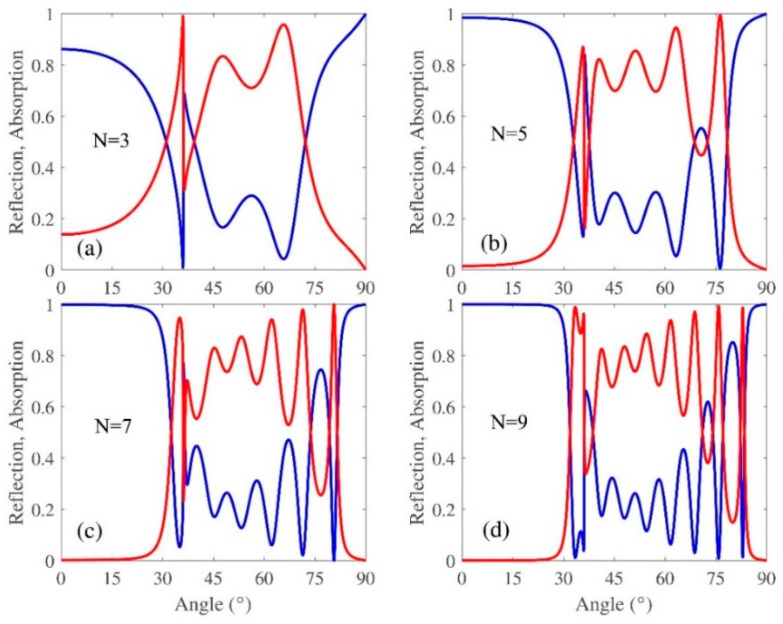
Absorption and reflection spectra of the DBR conjugated with the Bi_2_Se_3_ layer at a different period number of TiO_2_/SiO_2_ Bragg mirror under TM polarization. (**a**) *N* = 3, (**b**) *N* = 5, (**c**) *N* = 7 and (**d**) *N* = 9. The other parameters are the same as in Figure 3.

**Figure 5 micromachines-13-01020-f005:**
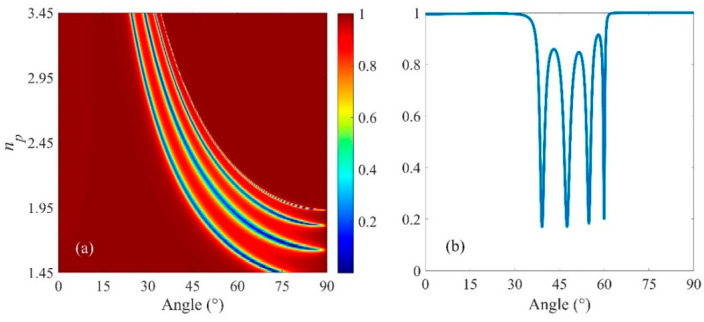
(**a**) Reflectivity spectra as a function of *n_p_* and the incident angle with *N* = 5, *d*_1_ = 96 nm, *d*_2_ = 172 nm and *t* = 80 nm. (**b**) Simulated reflection spectra of the planar structure with *n_p_* = 2.45.

**Figure 6 micromachines-13-01020-f006:**
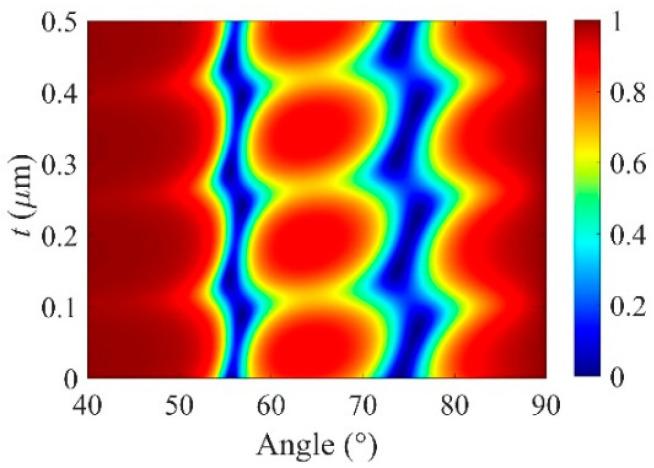
Reflectivity map on the plane of the incident angle and *t* with *n_p_* = 1.7. The other parameters are the same as used in Figure 3.

**Figure 7 micromachines-13-01020-f007:**
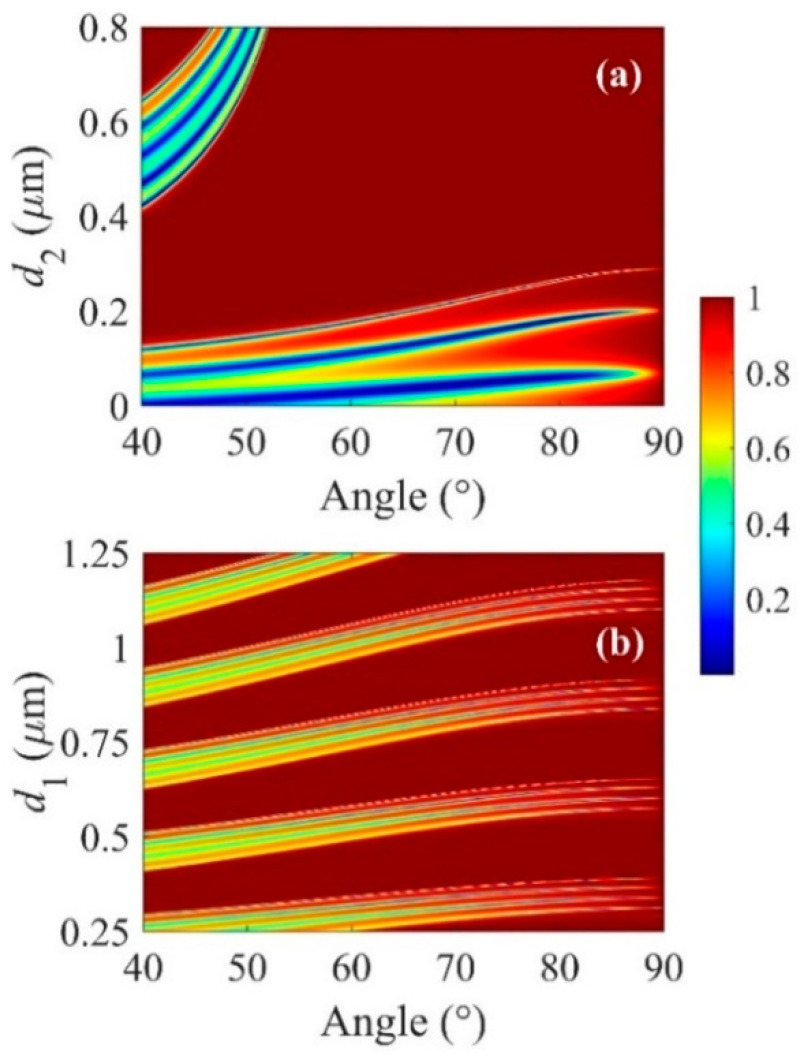
Contour map of reflection spectra as a function of (**a**) *d*_2_ and (**b**) *d*_1_.

**Figure 8 micromachines-13-01020-f008:**
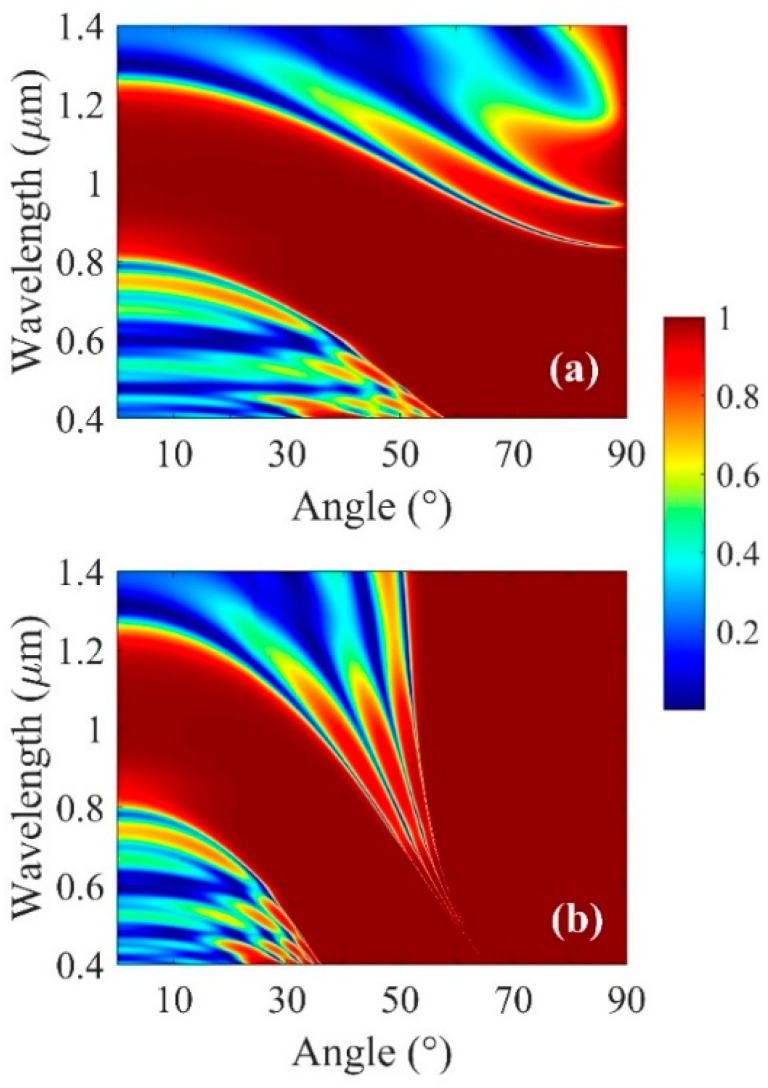
Evolution of absorption spectrum from the multilayer structure with the light incident angle *θ* when (**a**) *n*_p_ = 1.7 and (**b**) *n*_p_ = 2.45, respectively.

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
