# Peer review of "Excitation of Hybrid Waveguide-Bloch Surface States with Bi2Se3 Plasmonic Material in the Near-Infrared Range"

_micromachines, 2022, doi:10.3390/mi13071020_

Round 1
Reviewer 1 Report
The manuscript entitled "Excitation of hybrid waveguide-Bloch surface states with Bi2Se3 topological insulator in the near-infrared range" demonstrates the excitation of BSW using a Bi2Se3-DBR composite. The BSW's behavious are also studied by adjusting the light's initial states and material properties and geometries of the composite. I want to give the following comments:
(#1) This manuscript's title contains 'topological insulator'. But I cannot not find any behaviors related to the topological physics. To mention the topological insulator, the authors should show (1) band degeneracies, like Dirac/Weyl points or nodal lines, or (2) topological quantities, like Chern numbers or Berry phases of the photonic bands, or (3) specific phenomena, like surface states related to the nontrivial topological bulk states. If the results in the manuscript are not related at least one of the above, the Bi2Se3 composite was not used as a topological insulator but just a material. So, I suggest (a) proposing at lest one of the above or (b) removing the expressions 'topological insulator' in the title and main text.
(#2) To my knowledge, there have large amount of researches on Bi2Se3 materials. Even if you limit them as Bi2Se3 topological insulators, there are sill many works. Then, there will be a starting or big paper (drastically cited) that firstly proposed Bi2Se3 topological insulators. I think the paper(s) should be also cited in this manuscript.
(#3) In the abstract and main text, the expression 'purely dielectric' is used. But the meaning is unclear. Sometimes purely dielectric system refers a system whose permittivity is completly constant. So, I wonder if you used 'purely dielectric' as this meaning. Furthermore, as shown in Eq. (1) and (2), you system's refractive indices are function of a wavelength. Thus, you should logically clarify how your design has the advantage in terms of purely dielectric system. If not, removing 'purely dielectric system' is another method to improve the manuscript.
(#4) When I read a paper, first I search 'however paragraph'. From the paragraph starting with 'However, ......', I can easily know the significance of the paper. However, this paper does not strongly show the however paragraph although there is a sentence 'but it is barely reported ......'. How about strengthen this paragraph by reorganizing the second paragraph or adding one or two more sentence supporting the sentence 'but it is ......'.
(#miscellaneous comments 1) The manuscript's English style is a little bit old. How about getting English editage?
(#miscellaneous comments 2) Eq. (1) and (2) have lambda. But the meaning of the lambda is missing. (Interestingly the meaning appears in the first paragraph of page 6.)
Author Response
- The reviewer indicates that the Bi2Se3 composite was not used as a topological insulator but just a material. And the suggestion of proposing at least one of the topological properties or removing the expressions 'topological insulator' in the title and main text.
Reply: Following the suggestion of the reviewer, we have removed the expressions 'topological insulator' in the title and main text. Even Bi2Se3 composite owns topological properties, it is just used as one of the plasmonic materials in our study.
- The reviewer indicates the citation of the starting paper of Bi2Se3 topological insulators.
Reply: Thanks for your kind reminder. The paper Nature Phys. 5, 438–442 (2009) has been cited in the revision.
- The reviewer indicates the removing of 'purely dielectric system' in the manuscript.
Reply: Thanks for your suggestion. We are sorry for the misunderstanding of the statement about “purely dielectric system”. We have changed this point in the revision.
- The reviewer indicates the strengthen of the second paragraph.
Reply: This is a valuable suggestion. So far, most of the research are concerned on realizing narrow and tunable resonance based on BSW, but it is barely reported that the tunable and multichannel BSWs can be achieved simultaneously. Moreover, a novel Bi2Se3 platform the excitation of BSWs is significant for two-dimensional integrated photonics.
- We have added the meaning of λ and tried our best to improve the language.

Reviewer 2 Report
This paper demonstrates the possibility of using Bi2Se3-DBR composite structure for excitation of Bloch surface wave with advantages in terms of practical applications. The study is complete and very interesting, but I suggest in the introduction and conclusions paragraphs to precise the applications that can exploit the obtained structure.
Author Response
- The reviewer indicates “I suggest in the introduction and conclusions paragraphs to precise the applications that can exploit the obtained structure”.
Reply: The extraordinary properties of Bloch surface waves (BSWs) have made them an attractive spot in fundamental research and further applications, including non-specular reflection phenomenon optical sensing, optical switch, Raman scattering and so on. BSWs are considered a promising alternative to surface plasmon polaritons in optical sensing applications. The proposed structures in our work may not only find application as efficient narrowband filters as well as beam splitters and deflectors, but also extend the usage of Bi2Se3.
In summary, we have made every effort to improve our manuscript and match the comments from the reviewer’s report. We hope that this revised manuscript can be accepted for publication in the micromachines.
